

# Geophysical fingerprint of the 4-11 July 2024 eruptive activity at Stromboli volcano, Italy.

Luciano Zuccarello[1,2], Duccio Gheri[1], Silvio De Angelis[2,1], Riccardo Civico[3], Tullio Ricci[3], Piergiorgio Scarlato[3], Massimo Orazi[4].

1Istituto Nazionale di Geofisica e Vulcanologia, Sezione di Pisa, via Cesare Battisti, 53, 56125, Pisa, Italy.
2School of Environmental Sciences, University of Liverpool, 4 Brownlow Street, L69 3GP, Liverpool, UK.
3Istituto Nazionale di Geofisica e Vulcanologia, Sezione di Roma1, via di Vigna Murata, 605, 00143, Roma, Italy.
4Istituto Nazionale di Geofisica e Vulcanologia, Osservatorio Vesuviano, via Diocleziano, 328, 80124, Napoli, Italy.

*Correspondence to*: Luciano Zuccarello (luciano.zuccarello@ingv.it); Duccio Gheri (duccio.gheri@ingv.it)

**Abstract.** Paroxysmal eruptions, characterized by sudden and vigorous explosive activity, are common events at many open-vent volcanoes. Stromboli volcano, Italy, is well-known for its nearly continuous degassing activity and mild explosions from the summit craters, occasionally punctuated by energetic, short-lived paroxysms. Here, we analyse multi-parameter geophysical data recorded at Stromboli in early July 2024, during activity that led to a paroxysmal eruption on 11 July. We use seismic, infrasound and ground deformation data, complemented by visual and Unoccupied Aircraft System observations, to identify key geophysical precursors to the explosive activity and reconstruct the sequence of events. Elevated levels of volcanic tremor and Very Long Period (VLP) seismicity accompanied moderate explosive activity, lava emission and small collapses from the north crater, leading to a major explosion on 4 July, 2024 at 12:16 (UTC). Collapse activity from the North crater area continued throughout July 7, while effusive activity occurred from two closely-spaced vents located on the Sciara del Fuoco slope, on the Northwest flank of the volcano. On 11 July, a rapid increase in ground deformation preceded, by approximately 10 minutes, a paroxysmal event at 12:08 (UTC); the explosion produced a 5 km-high eruptive column and pyroclastic density currents along Sciara del Fuoco. We infer that the early activity in July was linked to eruption of resident magma within the shallowest parts of the volcano plumbing. This was followed by lowering of the magma level within the conduit system as indicated by the location of newly opened effusive vents The rapid inflation observed before the paroxysmal explosion on 11 July is consistent with the rapid expansion of gas-rich magma rising from depth, as frequently suggested at Stromboli during energetic explosive events. Our results provide additional valuable insights into the eruptive dynamics of Stromboli and other open-conduit volcanoes, and emphasize the importance of integrated geophysical observations for understanding eruption dynamics, their forecasting and associated risk mitigation.

## 1 Introduction

Stromboli is an open conduit stratovolcano located in the Tyrrhenian Sea, off the northern coast of Sicily; its activity is characterized by continuous degassing and frequent, small-to-moderate, explosions occurring every few minutes from the





summit craters, the well-known Strombolian activity. However, activity at Stromboli can rapidly escalate into more energetic
events, referred to as major explosions, which eject centimeter-to-meter-sized ballistic projectiles; at times, sustained explosive
activity is accompanied by partial collapses of the crater rim due to the instability of accumulated material, and increased
magmastatic pressure within the conduit system (Gurioli et al., 2013; Di Traglia et al., 2024). Since 2019, major explosions at
Stromboli have occurred with a frequency of about 4-5 events per year ejecting pyroclastic material to heights over a hundred
meters, which can travel beyond the summit crater area and potentially affect tourist paths (Rosi et al., 2013; Gurioli et al.,
2013). In heightened states of activity, Stromboli may also experience paroxysms, that is highly energetic eruptions that
generate eruptive columns exceeding 4 km in height, ballistics of up to 2 m in diameter and significant collapse activity from
the summit crater areas (Fig. 1). Paroxysms can be accompanied by the emplacement of pyroclastic density currents (PDCs)
along the Sciara del Fuoco (SdF, Fig. 1a), which can enter the sea and travel up to 2 km from the shoreline with demonstrated
potential to trigger tsunamis (Rosi et al., 2006; Calvari et al., 2006; D'Auria et al., 2006; Ripepe and Lacanna, 2024). Although
paroxysms are less frequent than major explosions, with an average occurrence of just one every four years since 2003, they
are the most impactful hazard for the island of Stromboli (Rosi et al., 2013). A recent paroxysm on 3 July, 2019, resulted in a
fatality (Giudicepietro et al., 2020; Giordano and De Astis, 2020; Andronico et al., 2021).
Unrest and eruption at Stromboli generate a broad range of geophysical signals. Nucleation and coalescence of gas bubbles
into gas slugs (Sparks, 2003; Burton et al., 2007; Caricchi et al., 2024), and their ascent within the conduit generates
characteristic seismic and deformation signals (Marchetti et al., 2009); gas slug bursting at the top of the magma column
produces infrasound waves (Colò et al., 2010). Real-time detection and monitoring of these signals are crucial for risk
mitigation at Stromboli as, in the recent past, major explosions and paroxysms have frequently been anticipated by detectable
changes in geophysical signals between tens of seconds and minutes before their occurrence (Giudicepietro et al., 2020; Ripepe
et al., 2021a; Longo et al., 2024).
Except for the 2019 eruptive activity, the most intense in recent years, Stromboli's paroxysms are typically preceded by periods
of lava effusion, or a general increase in surface activity that lasts for several days (Ripepe et al., 2009; Valade et al., 2016).
Several studies have suggested that effusive eruptions may act as a trigger for paroxysmal explosions through a mechanism of
decompression of the volcano plumbing system, evidenced by a drop in magma levels within the conduit (Aiuppa et al., 2010;
Calvari et al., 2011; Ripepe et al., 2017). The most significant effusive event in terms of its volume occurred between December
2002 and July 2003 (Ripepe et al., 2017), which caused landslides, triggered a partial collapse of the SdF and culminated in a
paroxysm on 5 April, 2003; this was the first large-scale paroxysmal event on record since 1985 (Calvari and Nunnari, 2023).
However, it should also be noted that effusive eruptions are not necessarily followed by paroxysms. An example is the
November 2014 effusive eruption, which did not lead to paroxysmal activity (Rizzo et al., 2015). At the other end of the
spectrum lies the paroxysm of July 2019, for which no clear increase in activity prior to the main event was recorded. As
highlighted by Laiolo et al. (2022), thermal and gas flow levels had slightly increased but remained below "alert" thresholds.
Multi-parameter data are crucial to understand unrest at Stromboli and to detect transitions between low-to-moderate activity
and more explosive phases (Pistolesi et al., 2011; Andronico et al., 2021). Several conceptual models have been proposed



accounting for the ordinary seismic activity observed at Stromboli and other similar volcanoes (e.g., Chouet et al., 2008;
Suckale et al., 2016; Ripepe et al., 2021b). Petrological analyses of erupted products suggest the presence of a stratified conduit
at Stromboli, consisting of two types of magma (Bertagnini et al., 2003; Francalanci et al., 2004; Francalanci et al., 2005). The
upper conduit is thought to host highly porphyritic (HP) magma that is water-poor and rich in phenocrysts, and is erupted as
scoria during ordinary activity; on the other hand, magma in the lower conduit is gas-rich, low-porphyritic (LP), and typically
erupted as pumice alongside HP scoria and lithic blocks removed from conduit walls. Eruptive activity at Stromboli is inferred
to be controlled by the buoyant ascent and bursting of gas slugs (Sparks, 2003; Burton et al., 2007; Caricchi et al., 2024;
Aiuppa et al., 2010) from the top of the LP magma, rising through the more crystalline HP magma acting like a viscous fluid
or a rigid plug and controlling the final ascent and explosion of the slugs (Suckale et al., 2016). A recent model by (Caricchi
et al., 2024) shows that the instability of gas-rich and low-density foam layers at the base of the magma column could also
potentially trigger paroxysmal explosions at open conduit volcanoes.
In this study, we report on the most recent paroxysm at Stromboli, which occurred on 11 July, 2024, after a month of unrest at
the summit craters, as reported by the Istituto Nazionale di Geofisica e Vulcanologia (INGV) (INGV-OE, 2024). We analyze
the precursory geophysical activity leading up to the paroxysm based on seismic, infrasound and ground deformation data
gathered by the INGV monitoring network, complemented by observations conducted with an Unoccupied Aircraft System
(UAS) during the study period. The UAS imagery provides a valuable tool to interpret geophysical data and understand the
conditions leading up to the paroxysm on 11 July, offering a high-resolution reconstruction of the eruptive events and
associated morphological changes at the volcano. Unless, otherwise stated, all descriptions of surface activity in this
manuscript are from direct field observations by the authors during the study period.

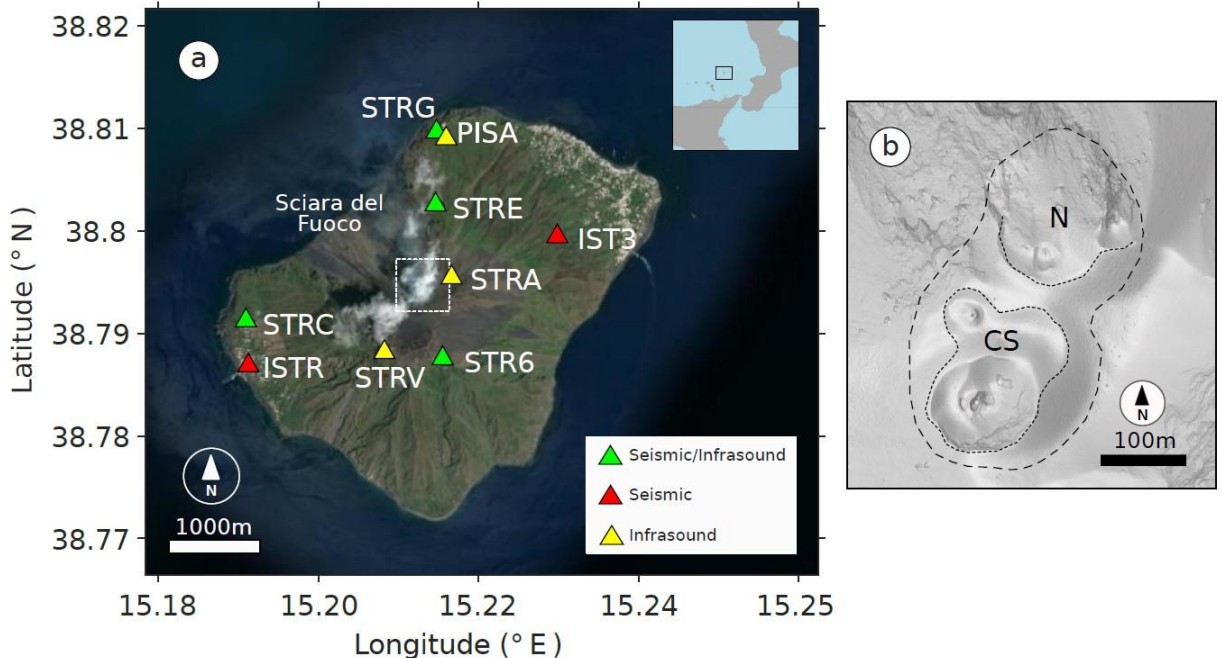




**Figure 1: a) Map of monitoring network at Stromboli, showing the locations of seismo-acoustic, seismic, and infrasound sensors. The inset shows the location of Stromboli volcano in Italy (MATLAB-Mapping Toolbox). b) Detail of the summit area of Stromboli, corresponding to the white dash-line square in a), showing the summit crater areas.**

## 2 Chronology of eruptive activity during 3-11, July, 2024

The activity bulletins issued by INGV (see Data Availability), from May 24 until the early days of July, reported an increase in surface activity at Stromboli, particularly from the North (N) crater area (Fig. 1b), characterized by continuous and intense spattering, that is quasi-continuous emission of pyroclastic material through sequential, small-to-moderate, explosions ejecting ballistics at heights of ~10-20 m above the vent (Harris and Ripepe, 2007; Giudicedipietro et al., 2021) (Fig. 2a). The average frequency of explosions fluctuated between 13 (medium) and 16 (high) events/hour with spattering occasionally leading to lava flows along the SdF (Fig. 1a). On June 23 and 28, lava flows began, following intense spattering from the N crater, converging into a canyon-like structure created by previous PDC activity in October 2022 (Di Traglia et al., 2024). Sulfur dioxide ($SO_2$) and carbon dioxide ($CO_2$) emissions remained at average levels, as did the carbon-to-sulfur (C/S) ratio (INGV-OE, 2024).

On 3 July, at 16:35 UTC, intense spattering was observed from a vent located within the N crater sector, leading to a sequence of partial collapses of the N crater rim, which also remobilized material that had been erupted in the preceding days. These collapses mostly consisted of cold material with a minor contribution of hot deposits. At 17:02 UTC, a lava flow began from the same vent, accompanied by spattering and moderate explosions (Fig. 2b). The activity continued throughout the night, with lava fronts moving down to an elevation of 550-600 m a.s.l..

On 4 July, at 12:16 UTC, a major explosion occurred from the N crater and, at 14:10 UTC, a new lava flow emerged at the base of the N crater area at ~700 m a.s.l., advancing towards Bastimento and Filo di Fuoco, located along the northeast boundary of SdF. After about one hour a second lava flow started at an elevation of ~580 m a.s.l., which reached the sea. At 16:15 UTC, another vent opened at ~510 m a.s.l., producing a third lava flow accompanied by PDCs that rapidly descended the SdF into the sea (Fig. 2c). During the evening of 4 July, and throughout the following night, lava flow activity continued, accompanied by occasional collapses of pyroclastic materials.

Between 5-6 July, 83 landslide events were observed, while effusive activity fluctuated and lava emission moved further downslope originating from two new eruptive vents at ~485 m a.s.l. (Fig. 2d). The flow formed a delta at the shoreline and steam plumes were observed caused by magma-seawater interaction. Explosive activity from the summit craters halted at the beginning of the effusive phase.

On 11 July, at 12:08 UTC, a paroxysmal eruption occurred from the N crater area, producing an ash plume mn ~5 km high, which dispersed towards the southwest (Fig. 2e). Shortly after, a pyroclastic flow rapidly advanced along the SdF, which triggered a small-scale tsunami wave. The paroxysmal phase ended with a series of secondary and less intense PDCs.

In the following hours, effusive activity ceased, and no further explosions were observed, except for a minor event on 12 July, at 08:28 UTC (Fig. 2a), which was followed by a small collapse event in the N crater area.



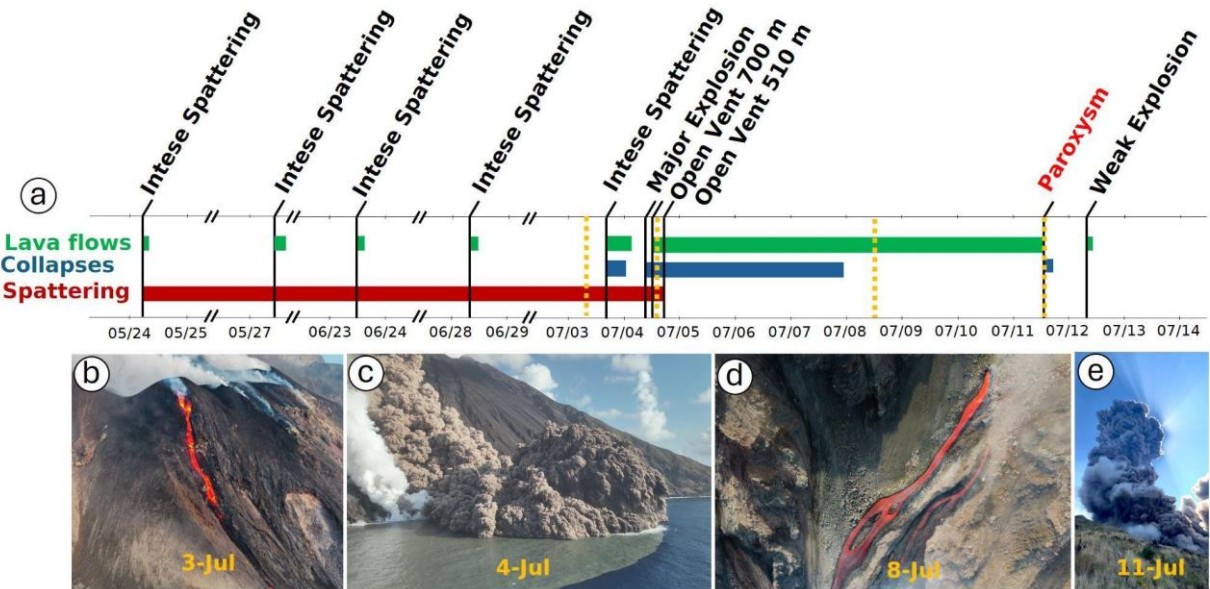

**Figure 2: Timeline of the observed surface activity and key visual observations at Stromboli between late May and mid-July, 2024.**
**a) Timeline showing the chronology of activity, which marks periods of activity characterized by lava flows (green), collapses (blue)**
**and spattering (red). Significant events are labelled, such as intense spattering, a major explosion on 4, July, opening of new vents,**
**and the paroxysm on 11, July. b-e) Sequence of images gathered at the times indicated by the dashed yellow lines in a). From left to**
**right: spattering activity on 3, July, a PDC event reaching into the sea on 4, July, continued lava flow on 8, July, and the paroxysmal**
**explosion on 11, July (photo "e" courtesy of G. De Rosa - OGS).**

## 3 Geophysical observations

In this study we use data recorded by the geophysical monitoring network deployed and maintained on Stromboli by INGV
(Fig. 2a). The network includes seismic (ISTR3, ISTR) and infrasound sensors (STRA, STRV), as well as seismo-acoustic
stations (STR6, STRC, STRE, STRG). An additional infrasound sensor, PISA (Gheri et al., 2024), was deployed on 4 July at
13:35 UTC, 35 minutes before the onset of the effusive activity.

### 3.1 Seismic characterization of unrest and eruptive events

Volcanic tremor is traditionally thought to reflect magma movement within the conduit (McNutt and Nishimura, 2008; Chouet
et al., 1997; Ripepe and Gordeev, 1999); at Stromboli, tremor is routinely monitored by means of the Root Mean Square (RMS)
of the continuous seismic signal in the 1-3 Hz frequency band (Giudicepietro et al., 2023). Figure 3a shows RMS values of the
order of $10^{-6}$ ms$^{-1}$ (recorded at the IST3 site), which correspond to tremor classified by INGV as high. A marked and short-
lived increase in seismic RMS was observed after the major explosion at 12:11 on 4 July (Fig. 3a). During this period, the
signal reached unprecedented levels, peaking at $10^{-4}$ m s$^{-1}$ at 17:00 UTC. Short-lived increases in RMS values were still noted



throughout 5, July, although the amplitudes exhibited an overall decline to values of the order of 10-7 m s-1, lower than those
recorded at the beginning of July. In the following days (6-11, July), the tremor was marked by a series of short-duration peaks
during lava flow activity. This behavior changed again on 11, July, when the onset of paroxysmal activity coincided with a
new increase in RMS (Fig. 3a). After the paroxysm, the RMS decreased again with only sparse and brief intervals of increased
amplitudes between 12-13, July (Fig. 3a). From late on 13 July, onwards, the amplitude stabilized around 10-7 m s-1, indicating
that volcanic activity had reduced and returned to background levels. Additional details of the signals recorded on 4-7 July,
are shown in the Supplementary Materials (Fig. 1S).
The spectrogram in Fig. 3b shows nearly continuous energy in the 2-3 Hz range, typically associated with tremor signals at
Stromboli (Ripepe et al., 1996). Energy levels in this band change throughout the pre-, syn-, and post-explosive activity
periods, reaching a maximum on 4 July following the major explosion. A pulsating phase was observed from 6-11 July, with
another peak during the paroxysm. Explosive activity between 4-11, July, exhibited a broader frequency range in the 0.5-15
Hz band. It is worth noting that the eruptive event on 4, July was preceded by a high-energy signal in the narrow frequency
band 0.2-0.3 Hz (Fig. 3b). We also observe that this very low-frequency signal was not recorded before the paroxysm on 11,
July.  Finally, on July 10 at 05:09 UTC and on 11 July at 02:26 and 15:21 UTC, high-energy signals were observed around
0.05-0.08 Hz, exhibiting a dispersive spectrum typical of teleseismic events as reported by USGS (for further information, see:
https://earthquake.usgs.gov/earthquakes/search/).
We have also analysed the occurrence of Very Long Period (VLP) earthquakes that have traditionally been associated with
pressure disturbances and the dynamics of gas-rich magma within fluid-filled structures (Chouet et al., 1997; Chouet et al.,
1999; Marchetti and Ripepe, 2005; Legrand and Perton, 2022), and one of the main tools used to monitor unrest at Stromboli.
An increase in the frequency of occurrence of these signals is typically a precursor to periods of elevated eruptive activity
(Ripepe et al 2009; Delle Donne et al., 2017). Figure 4a derived from information sourced from the INGV bulletins (INGV-
OE, 2024), provides an overview of the rates of VLP seismicity at Stromboli between the end of May and mid-July 2024, after
the 11 July paroxysm. From May until mid-June, VLP event rates remained stable, fluctuating around high values between 12
and 19 events/hour. A mean rate of ~13 events/hour is defined, at Stromboli, as "normal activity" (Ripepe et al., 2008) and it
suggests that an efficient degassing mechanism of the magma column is established (Ripepe et al., 2021b). A significant peak
is observed around mid-June, with the number of VLP events reaching a high of 19 events/hour on June 16. This peak is
followed by a slight decrease in event rates, although the number of events remained elevated compared to previous days.
Figure 4b shows the characteristic compression-decompression cycle of VLP events at Stromboli; this waveform represents
the normalized stack of all VLP events with maximum amplitude greater than 5 x 10-6 m s-1 at station STRE. Figure 4c shows
a 1-day filtered (0.03-3Hz) seismic record illustrating the occurrence of VLP events as recorded at station STRE, on the east
flank of SdF at 495 m of elevation (see Fig. 1).
Before the major explosion on 4 July, we observed a clear drop in the occurrence of VLP events (Fig. 4a) from 10-15 to 7-10
events/hour. The rates of VLP events remained stable until the 11 July paroxysm, peaking again at 12 events/hour on that day.
After the paroxysm, a further decrease in VLP rates was observed with hourly counts ranging from 6 to 10 events.





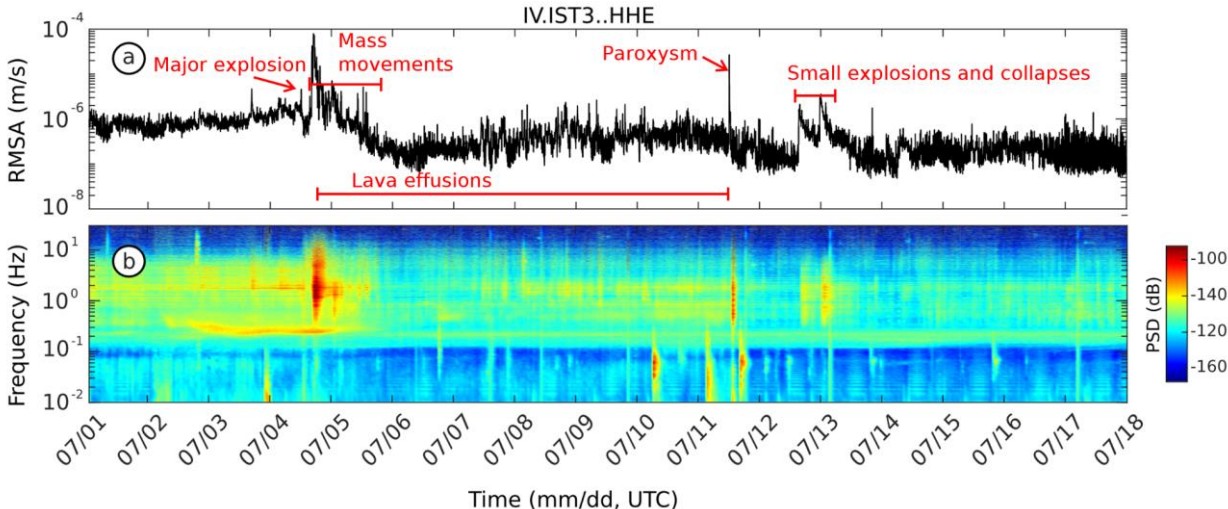

**Figure 3: a) Seismic tremor or RMS calculated every minute using a moving time window of 5 minutes, within the volcanic tremor frequency band of Stromboli (1-3 Hz), from July 2 to 18. b) Spectrogram of the E-component from the IST3 seismic station for the same period.**

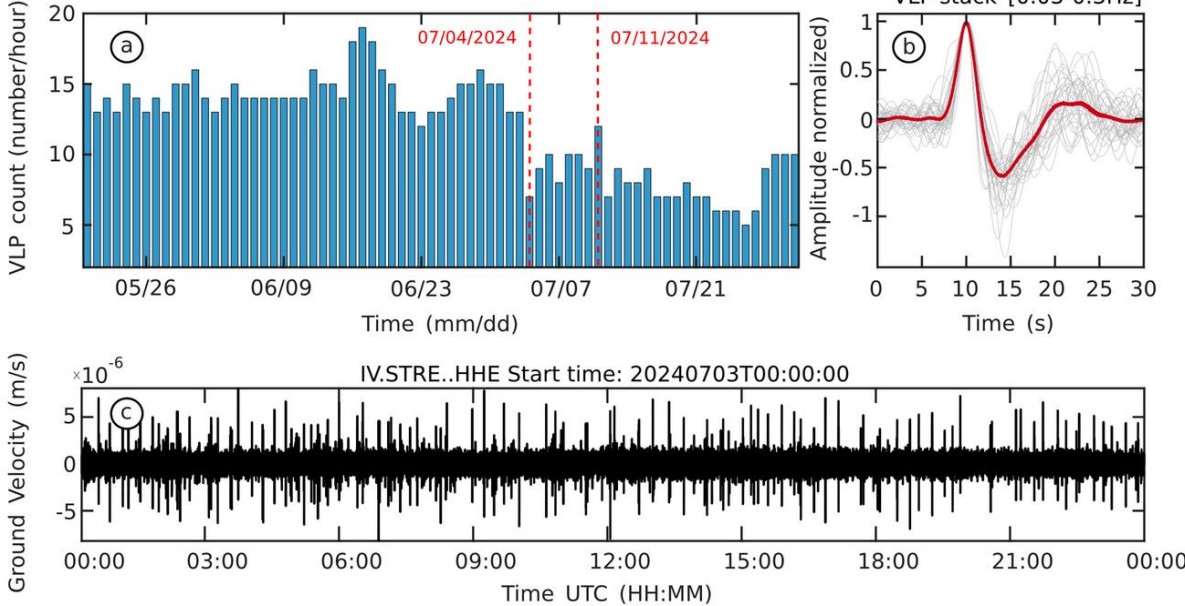

**Figure 4: a) Hourly rates of VLP events from the INGV catalog. Vertical red dashed lines indicate the major explosion and paroxysm that occurred on 4 and 11, July, respectively. b) VLP waveform events (>5×10-6 m s-1) recorded on 3, July, at station STRE normalized with respect to maximum amplitude (light grey). The red waveform represents the average of all high-amplitude waveforms. c) Continuous waveform recorded at station STRE (EW component) on 3, July 2024, filtered between 0.03-0.3 Hz.**



## 3.2 Infrasound characterization of unrest and eruptive events

We have also analysed infrasound data recorded by the INGV acoustic monitoring network and an additional microphone installed during the period of activity (Fig. 1). The infrasonic record before 4, July, shows a typical background of moderate strombolian activity occasionally interspersed with larger explosions (see Fig. 2Sa). The major explosion on 4 July, generated an infrasonic transient with a pressure of 5 Pa (Fig. 2Sb) at station STR6, from the CS crater area. Following this event, a marked decrease in acoustic energy was observed until the 11, July paroxysmal event, which produced infrasonic waves with a peak amplitude of 115 Pa at the STR6 site (approximately at ~750 m, see Fig.1a and Fig.2Sb).

We have used the infrasound records from all operating sensors of the INGV monitoring network on Stromboli and an additional temporary microphone (Fig. 2) to locate the source of the paroxysmal eruption on 11, July 2024. We employed the RTM-FDTD (Reverse Time Migration - Finite Difference Time Domain) method of Fee et al. (2021), which implements waveform back-projection over a grid of candidate source locations. Travel-times between potential source locations and all stations in the network are calculated via FDTD modeling (Kim and Lees, 2014; Fee et al., 2017; Diaz-Moreno et al., 2019) to account for the effect of topography on the propagation of the acoustic wavefield. In the RTM-FDTD method, waveforms are back-projected and a detector function (e.g., network stack, network semblance) is evaluated for each candidate source, with the detector maximum corresponding to the most likely location. For FDTD calculations of travel-times we employed a UAS-derived Digital Elevation Model (DEM) of the SdF and the summit craters (Civico et al., 2024) areas conducted on the morning of 4 July with initial individual resolutions ranging between 20 and 50 cm/pixel. This DEM was merged with a basemap elevation model (Civico et al., 2021) of the rest of the island, re-sampled, and parsed into a 5x5 m grid for the purpose of FDTD modeling. For FDTD modeling, the source time function was approximated by a Blackman-Harris function with a cutoff frequency of 5 Hz (high enough to include the dominant frequency of the explosion signals while still allowing time-efficient computing) and the acoustic wavefield was propagated along the discretized topography using 15 grid points per wavelength (Wang, 1996). We used a constant sound velocity of 330 m s-1 (estimated from the signal move-out across the network) and a stratified atmosphere model based on density and temperature data obtained from the Reanalysis v5 (ERA5) dataset (see Data and Resources), produced by the European Centre for Medium-Range Weather Forecasts of the Copernicus Climate Change Service. We used data corresponding to the ERA5 grid node closest to Stromboli, at 12:00 on 11, July 2024 (Coordinated Universal Time, UTC). The inferred source location for the paroxysmal explosion on 11, July 2024, along with a record section of the infrasound waveforms used and the detector function, are shown in Fig. 5. The location identifies a source located approximately 50m below the rim of the N crater (Fig. 5a) at an elevation of ~685 m. The estimated origin time for the event is 12:08:52 UTC.





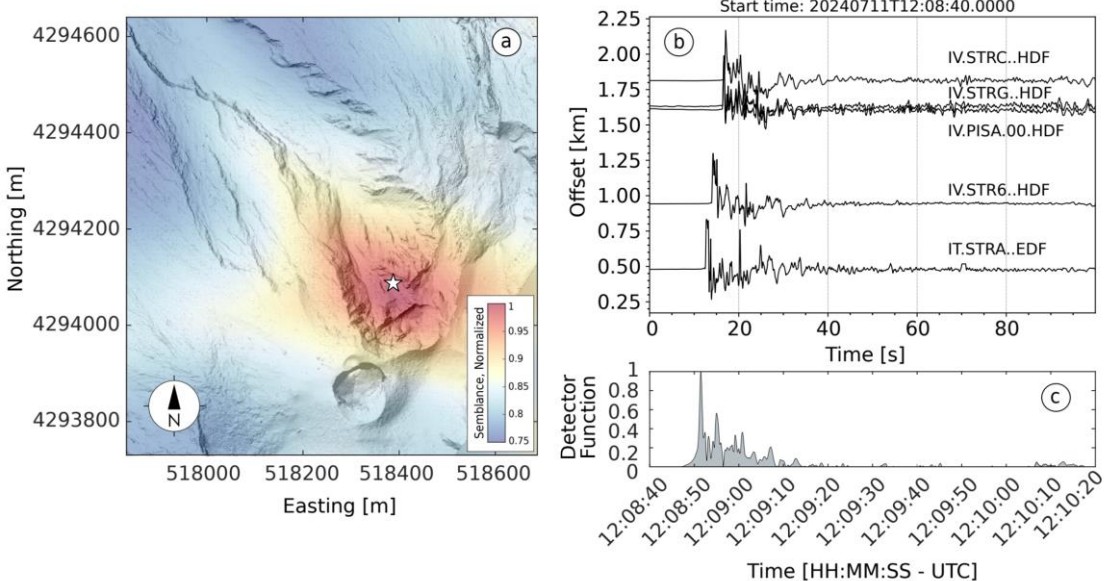

210

**Figure 5: Infrasound location of the 11 July, 2024 paroxysmal event using the RTM-FDTD method (see manuscript for details; DEM of July 14, 2024 from Civico et al. (2024). a) Map-view of network semblance maximum around the Stromboli crater region. RTM-FDTD semblance location is indicated by a white star; b) record section of the filtered infrasound waveforms (bandpass filter 0.01-15Hz) used for locating the event. The offset corresponds to source-station distance; c) Normalized network detector function (i.e., maximum network semblance amplitude over time).**

### 3.3 Deformation of unrest and eruptive events

Ground tilt at Stromboli has been frequently inferred to reflect processes like slug coalescence, slug ascent, and conduit emptying (Marchetti et al., 2009; Genco and Ripepe, 2010; Bonaccorso, 1998). Over the last decade, tilt has become central to real-time monitoring and eruption early warning at Stromboli. Ripepe et al. (2021a), for example, demonstrated the scale invariance of tilt at Stromboli, that is all explosions, regardless of their intensity, follow the same ground inflation-deflation pattern. A significant tilt was reported on 4 July (INGV-OE, 2024). The major explosion at 12:00 UTC was accompanied by a characteristic inflation-deflation pattern (Longo et al., 2024), followed by a pronounced deflation trend that began at 16:20 UTC and continued until 19:50 UTC (INGV-OE, 2024).

For the paroxysm on 11, July 2024 fig. 5 shows the seismic-derived tilt, reconstructed from the EW horizontal component record at station STRE Aoyama et al. (2008), Genco and Ripepe (2010), and De Angelis and Bodin (2012). Slow inflation is observed, starting approximately 600 seconds before the explosion (Fig. 5b); the seismic-derived tilt sharply accelerates approximately 1 minute before reaching its peak of 1.5 µrad at the onset of the explosion, followed by rapid deflation. This pattern is consistent with previous observations of tilt at Stromboli before paroxysms and major explosions (e.g. Genco and Ripepe (2010); Ripepe et al. (2021a)). We note that this tilt signal is derived from an individual seismic record, of an instrument



that is not likely oriented in the direction radial to the source; for this reason, we will focus on the interpretation of the
deformation trend, and will not use the measured tilt amplitude for modelling purposes.

## 4 Discussion

In this manuscript we have presented geophysical data recorded between early and mid-July 2024 at Stromboli; the period of
unrest included a major explosion on 4 July, significant collapse activity in the N summit crater area, emplacement of lava
flows, and a paroxysmal event on 11 July. Surface activity at Stromboli intensified late in May with a marked increase in the
occurrence of Strombolian explosions , the onset of effusive activity from SdF, and increasing volcanic tremor. Early in July,
we observed a steady increase in volcanic tremor reaching unprecedented amplitudes on 4 July, (see Fig. 3a and Fig. 1S).
Volcanic tremor at Stromboli has typically been linked to the coalescence of gas bubbles from layers of smaller bubbles and
their ascent along the shallower conduit (McNutt et al., 2008; Chouet et al., 1997; Ripepe et al., 1999), suggesting that
variations in tremor intensity are controlled by changes in gas flow within the conduit. It has been frequently speculated that
an increase in volcanic tremor reflects an increase in the volume of gas within the magma (Ripepe et al., 1996), which in turn
is linked to a higher occurrence of explosions at the top of the magma column. Field observations of increasing spattering in
early July (Fig. 1) support a model of increased surface activity linked to the ascent of gas-rich magma within the shallow
conduit. The high rates of VLP events observed during the same period further support the hypothesis of gas-rich magma
migration within the shallow plumbing system. These events are traditionally associated with the rapid expansion of gas slugs
rising through the liquid melt in the shallow conduit (Chouet et al., 2003; James et al., 2006); more recently (Ripepe et al.,
2021) suggested that VLP waveforms at Stromboli are generated at the top of the magma column, mainly after the onset of
Strombolian explosions; they showed that the occurrence of VLP event can be linked to explosive magma decompression in
the uppermost ~ 250 m of the conduit. The recorded VLP events showed similar waveforms (Fig. 4b) suggesting a stable source
mechanism and location; locations in the shallow parts of the conduit can be linked to magma accumulation at a shallow depth,
close to the surface. While the number of VLP events did not show any significant variation before the major explosion on 4
July, volcanic tremor increased slowly but steadily (Fig. 3a). Coinciding with strong ground deflation after the major explosion
(INGV-OE, 2024), volcanic tremor reached an unprecedented peak amplitude of ~8 x 10-5 m s-1 at ~17:00 UTC associated
with the opening of a new effusive vent at ~ 510 m elevation within SdF (Fig. 2a) and the occurrence of numerous mass
wasting events linked to collapse activity within the lower N crater area and upper section of SdF. We suggest that these signals
reflect the emptying of the shallowest parts of the conduit system and the overall lowering of the magma level within the
shallow volcano plumbing reflected in the opening of new effusive vents at progressively lower elevations. The transition
between explosive and effusive regimes was also marked by a clear decrease in the occurrence of VLP events (Fig. 4), and a
migration of their source deeper within the conduit (Ripepe et al., 2015). This contrasts with the flank eruptions of 2007 and
2014 (Ripepe et al., 2009; Ripepe et al., 2015) when VLP rates remained high during effusion; in July 2024 it appears that
effusion reduced the overall explosivity, rather than recalling fresh magma from depth. The new effusive regime, indeed, was



characterized by a substantial lack of Strombolian explosive activity at the surface between 4-11 July, as observed in the field
by our research team. The quasi-continuous collapse activity, observed from 13:00 UTC on 4, July, appeared to be linked to
instabilities in the crater area around newly created vents; this instability persisted in the following days, with the number of
events peaking on July 5 (83 recorded occurrences recorded in a single day (INGV-OE, 2024). The collapse activity recorded
along the N crater rim, adjacent to the SdF, resulted in significant changes to the morphology of this sector of the volcanic
edifice (Fig. 6).
During the study period, we also collected UAS data and compiled very high-resolution repeat DEMs (0.2-0.5 m/pixel), which
allowed quantifying topographical changes via DEM differencing. The difference between DEMs on 4, July, (morning) and
July 14 is shown in Fig. 7c. The data processing methodology follows the procedures described in Civico et al. (2022, 2024).
The most notable morphological variations were observed in the afternoon of 4 July, while the paroxysm on 11 July did not
lead to significant changes.
The summit craters were affected by loss of material due to the opening of two eruptive vents at approximately 700 and 500
m a.s.l.. While the CS crater sector showed a roughly circular-shape crater floor deepening of about 84 m, the N sector was
affected by the complete dismantling of its northern rim and external slope, marking the deepest morphological change
occurred at the summit craters in the last decades, with a maximum difference in altitude of 109 m. The total volume loss
recorded in the summit craters sector was estimated at 3.3 Mm3 (Civico et al., 2024).



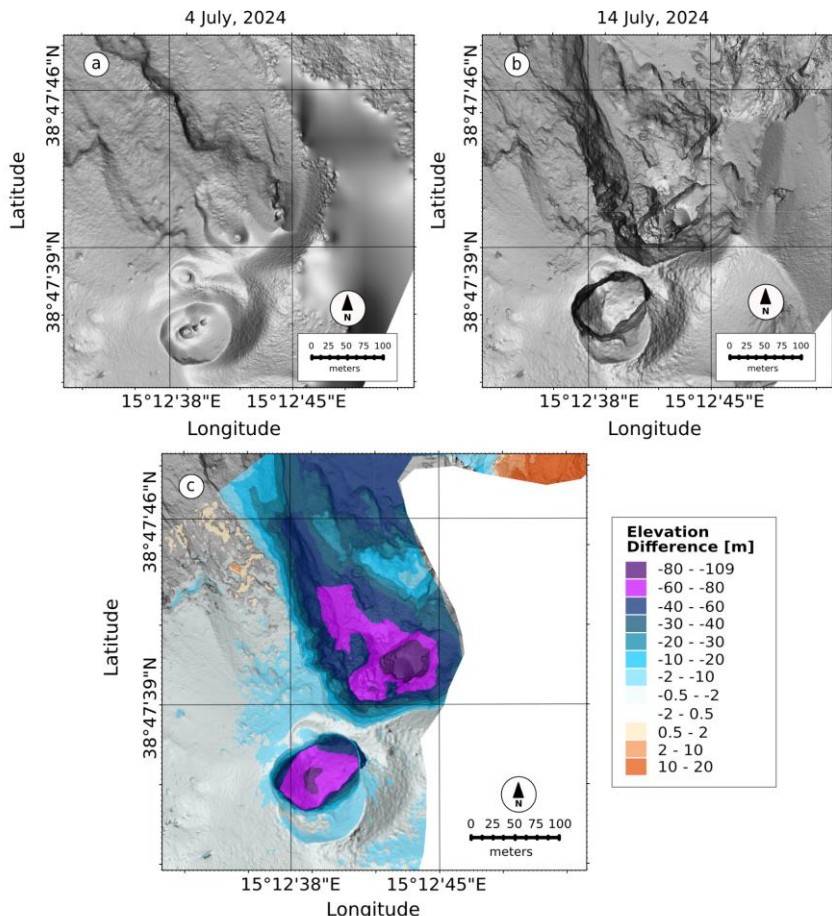

**Figure 6: Multidirectional hillshades of Stromboli's crater area: a) 4, July 2024 (Civico et al., 2024c), b) July 14, 2024 (Civico et al., 2024), c) map of elevation difference (Dem of Differences) highlighting morphological changes occurred between 4 and 14 July, 2024. Purple areas indicate material loss, whereas orange areas indicate material gain.**

Unlike the summit craters, the subaerial portion of the SdF slope was affected by both accumulation and erosion processes. Here, the main loss of material (2.74 Mm3; Civico et al., 2024a) was localized along the canyon formed in October 2022 (Di Traglia et al., 2024), which has widened and deepened during the July 2024 eruption. Accumulation processes instead were mainly due to PDC and lava flow deposits, localized in the northeastern sector of the slope. The maximum accumulation of lavas occurred at the new lava delta (maximum difference in altitude of 45 m), located in the center of the SdF shoreline.

The effusive regime ended with the occurrence of the paroxysmal explosion on 11, July. The explosion generated an infrasonic pressure of 115 Pa at station STR6 with an associated VLP amplitude reaching $5.8 \times 10^{-5}$ m s-1 (see Fig. 3S). An ash plume reached a height of 5 km above the vent, and pyroclastic flows moved down the SdF. After that, volcanic activity reduced its intensity, showing low levels of tremor and VLP events although the tremor increased again on 12, July, associated with a small lava flow.



The eruptive crisis of July 2024, culminating in the paroxysm, is consistent with previous eruptions at Stromboli, such as those
in April 2003, March 2007, and July-August 2019. The data discussed above can be used to inform a conceptual model of the
entire sequence of processes responsible for the observed surface and eruptive activity, within the framework of previous
studies (e.g., James et al., 2006; Chouet et al., 2008; Del Bello et al., 2012; Suckale et al., 2015; McKee et al., 2022).
The spattering activity, observed at the start of our study period, represents an intensified form of puffing. Spattering activity
results from the quasi-continuous bursting of small gas pockets within a bubbly flow regime, which generates pyroclasts
fragments (Rosi et al., 2013). This activity typically marks the initial stages of unrest and eruption at Stromboli, where gas-
rich magma is being actively degassed through continuous explosive bursts (Del Bello et al., 2012). At the more explosive end
of the spectrum of Strombolian activity major explosions and paroxysms are often explained invoking the "slug model" (James
et al., 2006; Chouet et al., 2008; Del Bello et al., 2012). In this model, gas bubbles (slugs) form deeper in the magma column
and gradually coalesce as they rise through the conduit due to an increase of the magma viscosity. As gas slugs ascend, they
expand due to decreasing pressure and eventually reach the surface. When they burst at the top of the magma column, they
release gas explosively, fragmenting the magma and producing pyroclasts and feeding ash plumes of varying sizes. After the
major explosion on 4 July, an effusive regime was established, characterized by lava flows, during which more degassed
magma was erupted. Following the initial explosive activity driven by gas slugs, we infer that the transition to effusive regime
is controlled by depressurization of the shallow plumbing system similar to Ripepe et al. (2017). The depressurization of the
system caused by the initial explosive activity allowed magma to flow, and reach the surface forming lava flows, without
further explosive activity. As the shallow volcanic conduit progressively emptied it leads to structural instability, causing
collapses and landslides along the SdF.
According to Ripepe et al. (2017), the emptying of the conduit creates a vacuum effect that draws more gas-rich magma from
deeper within the system. As volatile-rich magma rises and encounters lower pressures, it can lead to explosive eruptions,
resulting in a paroxysmal event. The dynamics of the 11, July paroxysmal explosion displayed similar trends across seismic,
acoustic, and deformation parameters compared to the others (Genco and Ripepe, 2010; Ripepe et al., 2021a). This consistency
further validates the established models of Strombolian activity, where the largest explosions and energetic events, such as
paroxysms, are driven by the same source mechanism. The scale-invariant conduit dynamics of ground deformation
demonstrate that inflation amplitude and duration scale directly with the magnitude of the explosion (Ripepe et al., 2021a).
Ground deformation observed on 11, July (Fig. 5) follows the same exponential inflation pattern as seen in previous paroxysms
(Ripepe et al., 2021a). This behavior is typically explained by bubble dynamics, where the pressure on the conduit walls
increases due to the rapid volumetric expansion of gas in highly vesiculated magma. As gas rises and expands, moreover, it
pushes the magma column toward the surface, often leading to precursory lava emissions from the vent. Ground deformation
is likely caused by a combination of increasing magma static pressure and the pressurization of degassed magma at the top of
the column, driven by the exponential growth of gas. When the pressure applied by the gas-rich magma exceeds the tensile
strength of the viscous magma plug, fragmentation occurs, resulting in the explosive release of gas and pyroclastic material
(e.g. paroxysm). Another possible mechanism, proposed by Suckale et al. (2016) and McKee et al., (2022) suggests that the



explosion is triggered by the rapid expansion and release of gas when a partial rupture occurs in the plug at the top of the
magma column.

## 5 Conclusion

The eruptive activity at Stromboli starting from 4, July, and culminating on 11, July 2024, with the paroxysm, provides a
comprehensive case study of explosive volcanism at open-conduit volcanoes, thus offering additional insights and proofs for
already existing source models.
The July 2024 paroxysm is preceded by a prolonged phase of heightened activity, characterized by increased volcanic tremor
and VLP events. The high seismicity, combined with observed crater rim collapses and lava flows, suggests a progressive
destabilization of the volcanic edifice. In particular, the major explosion on 4, July, and the subsequent paroxysm on 11, July
highlight the role of magma gas dynamics, where increased gas volumes and pressure led to significant eruptive events.
Seismic analysis reveals that the volcanic tremor intensity is linked to gas-rich magma movement, reaching in this eruptive
sequence unprecedented values at Stromboli. However, the variability in VLP events indicates that, while useful for monitoring
overall volcano unrest, these signals alone may not serve as reliable precursors for major explosive events. Instead, the
combined analysis of different geophysical parameters, including ground deformation, proved crucial for early warning and
forecasting as previously suggested by Ripepe et al. (2021a).
Ground deformation patterns, specifically the inflation-deflation cycle observed before explosions, align with previous studies,
confirming that such patterns reflect the occurrence of imminent explosions regardless of their magnitude. The exponential
inflation observed before the paroxysm, caused by gas expansion and the rise of slugs within the magma column, is the same
as in other paroxysmal events at Stromboli, supporting the already proposed source mechanism models for explosive events.
Through UAS data, Civico et al. (2024) were able to estimate a total volume loss of about 6.0 Mm3 involved after the
gravitational mass collapses occurred on 4 and 11 July. The partial collapses generated a reshaping of the summit craters area
as well as a deepening 2022 canyon along SdF, thus increasing the flank instability.
In conclusion, our results demonstrate how geophysical, visual observation and UAS-derived topographic data could offer
valuable insights for tracking the volcanic explosive phenomena as well as the partial collapses of the summit craters due to
the flank instability. This multiparametric monitoring approach could lead to significant advancements in reducing volcanic
hazards at Stromboli.

**Data availability**

The seismic waveform data from all the stations are part of INGV seismic network. The data are publicly available at EIDA
Italia (https://eida.ingv.it/). The infrasound data are available upon request from the INGV- Osservatorio Vesuviano. The
infrasonic collected from PISA station are available at https://doi.org/10.5281/zenodo.14245572.



**Author contribution**

L.Z., S.D.A. and P.S. wrote the proposals that funded installation and maintenance of the infrasound array and UAS, designed the field experiment, and financially supported this publication. L.Z. and S.D.A. tested the infrasonic equipment, organized fieldwork and participated in the original design of the experiment. L.Z., S.D.A., R.C., T.R. contributed to assembling the final multiparametric dataset and tested its quality and retrieval. L.Z., R.C. and T.R. installed and maintained the equipment during the field acquisition. M.O. managed and maintained the geophysics network, contributing to the seismo-acoustic data collection. L.Z, S.D.A. and D.G. performed analyses of infrasound data, seismic and tilt data, and prepared all figures. R.C. and T.R. analysed the UAS images. L.Z. D.G. and S.D.A. jointly wrote the initial draft of the manuscript and all authors contributed to review and edit the final version.

**Competing interests**

The authors declare that they have no conflict of interest.

**Acknowledgements**

INGV Project 'Pianeta Dinamico (Dynamic Planet) - Working Earth': Geosciences For The Understanding The Dynamics Of The Earth And The Consequent Natural Risks - "Dynamo - DYNAmics of eruptive phenoMena at basaltic vOlcanoes" (https://progetti.ingv.it/it/pian-din/dynamo#project-info).

INGV Departmental Strategic Project "UNO - UNderstanding the Ordinary to forecast the extraordinary: An integrated approach for studying and interpreting the explosive activity at Stromboli volcano" (https://progetti.ingv.it/it/uno-stromboli).

L.Z., D.G., S.D.A., R.C., T. R: and P.G. are supported by the grant "Progetto INGV Pianeta Dinamico" -Sub-project VT_DYNAMO 2023- code CUP D53J19000170001 - funded by Italian Ministry MIUR ("Fondo Finalizzato al rilancio degli investimenti delle amministrazioni centrali dello Stato e allo sviluppo del Paese", legge 145/2018).

We are indebted to all the colleagues who have contributed to the monitoring efforts on Stromboli during July 2024 and the ones involved in the surveillance and network maintenance activities, to Maria Zagari (Italian Civil Aviation Authority - ENAC) for her help in issuing new NOTAMs during the emergency, and to Giuseppe De Rosa, Istituto Nazionale di Oceanografia e di Geofisica Sperimentale (OGS) for providing the photo of the 11, July paroxysm in Fig. 1.

The contents of this article represent the authors' ideas and do not necessarily correspond to the official opinion and policies of the Dipartimento della Protezione Civile - Presidenza del Consiglio dei Ministri.e UAS images. We are grateful to the "Gruppo monitoring dell'Osservatorio Vesuviano" of INGV, Osservatorio Vesuviano (Italy), for their support in the data management.



**Financial support**

This work was supported by the grant "Progetto INGV Pianeta Dinamico" -Sub-project VT_DYNAMO 2023- code CUP D53J19000170001 - funded by Italian Ministry MIUR ("Fondo Finalizzato al rilancio degli investimenti delle amministrazioni centrali dello Stato e allo sviluppo del Paese", legge 145/2018) and by NGV Departmental Strategic Project "UNO - UNderstanding the Ordinary to forecast the extraordinary: An integrated approach for studying and interpreting the explosive activity at Stromboli volcano".

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
