# Peer review of "Geophysical fingerprint of the 4-11 July 2024 eruptive activity at Stromboli volcano, Italy."

_EGUsphere, 2024_

## Referee Comment (RC2)

[referee-annotated manuscript omitted]

---

## Author Response (AR1)

This document contains our responses to all reviewer comments, with corresponding references to the text. Additionally, we have attached the second reviewer's comments along with our point-by-point responses, as well as the revised manuscript with all changes highlighted in red.

**RC1**: 'Comment on egusphere-2024-3773', Anonymous Referee #1, 29 Dec 2024  reply

Luciano Zuccarello et al. provide a comprehensive analysis of the eruptive activity at Stromboli volcano from 3 to mid July 2024. The study utilizes a multi-parameter approach, incorporating seismic, infrasound, and ground deformation data, along with visual and UAS observations, to understand the dynamics of the eruption and its precursors. The study effectively integrates various geophysical data types, providing a holistic view of the eruptive processes.

This multi-parameter approach is crucial for understanding complex volcanic phenomena.

In the manuscript are offered:

1. A detailed chronology of events, from the initial signs of unrest to the paroxysmal eruption on 11 July. This timeline is well-supported by both observational data and geophysical measurements.

2. A clear presentation of the results, with well-constructed figures and tables that enhance the understanding of the data. The use of UAS imagery to complement geophysical data is particularly innovative and provides high-resolution insights into morphological changes.

3. Valuable insights into the eruptive dynamics of Stromboli and similar open-conduit volcanoes. It supports existing models of volcanic activity and highlights the importance of integrated monitoring for eruption forecasting and risk mitigation.

The authors should take into account -mainly for future work- a possible expansion of the discussion to include the broader implications of the findings. How can these insights be applied to other volcanic systems either in Italy or elsewhere? What are the potential impacts on volcanic hazard assessment and mitigation strategies in Arc Volcanoes in the Mediterranean region of similar environments globally?

Overall, the manuscript is a significant contribution to the field of volcanology, providing detailed and valuable insights into the eruptive activity at Stromboli volcano. The soundness of the methodology and the conclusions can be supported by the results, and therefore, I recommend this research paper to be published in its present form.

REPLY

Many thanks for these comments. We have slightly modified the discussion and conclusions to emphasize the vital importance of multiparametric monitoring on conduit volcanoes like Stromboli.

**Citation**: https://doi.org/10.5194/egusphere-2024-3773-RC1

**RC2**: 'Comment on egusphere-2024-3773'

The manuscript by Zuccarello et al. addresses a relevant and timely topic both scientifically and in terms of risk management, namely the comprehensive analysis of geophysical signals associated with the eruptive activity of V. Stromboli in July 2024. Added to this is the fact that such activity has been one of the most intense for this volcano in recent years, reaching unprecedented seismic energies and causing severe remodelling of the crater area and northern flank. The chronology and relationship of events and observations made by the authors is very complete and exhaustive. This work is innovative and original, integrating multiparametric data, both public and of their own acquisition, into a well-formulated, simple, clear, coherent methodology based on postulates and physical models appropriate for the object of study. The techniques used are not new or original, but the way of integrating the results and their interpretation for a recent and extraordinary eruptive event is valid and innovative, which is why it is of great interest to the scientific community.

This work completes and advances previous work by some of its authors, providing useful evidence for the conceptual and analytical modeling of the eruptive dynamics of V. Stromboli. The authors carry out a detailed recapitulation, analysis and interpretation of the events and data associated with the eruptive event, based on the state of the art in this matter. It is to be expected that the authors themselves or their collaborators will feed back the existing models on the dynamics of V. Stromboli with these conclusions and that they will effectively be incorporated into the elaboration of their eruptive forecasts.

Below, I detail some particular comments on the different sections and attach the pdf text with some added notes on highlighted lines that I hope can help to further improve this work.

As far as I can assess, the authors make good use of the language. The technical terminology is adequate and precise. A few typos and superscript errors should be corrected for the final version, as well as considering the unification of styles in the date formats used. The illustrations are clear, well defined, and with texts and colours that allow for the good interpretation. I have only left one comment about a figure that is mentioned in the text and has not been incorporated into the preprint. This detail has caused the references to some figures to be wrongly numbered (see comments in the text).

As for the structure of the manuscript, I consider it to be well-designed, ordering the information and its analysis in a coherent manner. The titles of the sections have been appropriately selected and their contents are generally well developed.

The abstract is concise and complete, allowing the objectives and results of the work to be understood in its entirety.

The introduction makes a very exhaustive compilation of the background that justifies and motivates this work and also introduces the reader to the knowledge base of V. Stromboli in an enjoyable way.

Section 2, whose title announces the eruptive chronology from July 3 to 11, begins by describing the activity from May 24. Perhaps the authors could change the title of this section.

Section 3, which refers to the geophysical observations, is too brief and could be expanded with some more detailed information about the instruments used, their characteristics, and the selection of the data for the analysis.

In section 3.1 I have left some comments and suggestions whose consideration would help the reader to better understand the authors' criteria in the analysis of the continuous tremor and the VLPs.

In section 3.2, in addition to verifying the values mentioned in the text and graphs for the frequency ranges used, I have found some differences between the Pa amplitudes in figures 2Sa, 2Sc, and the text.

Section 3.3 lacks the figure mentioned in the text.

The discussion is very rich and well-elaborated based on the results of the analyses of this work and its comparison with previous processes. The conclusions adequately bring together all the information provided, reaching a good synthesis of the results and their scope.

The selected bibliography brings together state-of-the-art on the subject and is generally well-referenced. I have left only a few comments in the text on this subject.

I consider that with some slight modifications to the text and the incorporation of the missing figure, this work is ready to be published and I strongly recommend its inclusion in NHESS

REPLY

Many thanks for these comments. We have updated the manuscript accordingly. We have addressed the major comments here, while in the attached PDF, we have responded to each individual comment directly within the text.

*Section 2, whose title announces the eruptive chronology from July 3 to 11, begins by describing the activity from May 24. Perhaps the authors could change the title of this section.*

Yes, we agree with you. We have modified the section title to make it more general: 'Chronology of Eruptive Activity'.

*Section 3, which refers to the geophysical observations, is too brief and could be expanded with some more detailed information about the instruments used, their characteristics, and the selection of the data for the analysis.*

We have described the INGV seismo-acoustic network used in this study at the beginning of Section 3.

*In section 3.1 I have left some comments and suggestions whose consideration would help the reader to better understand the authors' criteria in the analysis of the continuous tremor and the VLPs.*

We have added and rephrased some sentences to describe the nature of tremor and VLP (see specific replies in the attached PDF).

*In section 3.2, in addition to verifying the values mentioned in the text and graphs for the frequency ranges used, I have found some differences between the Pa amplitudes in figures 2Sa, 2Sc, and the text.*

Thank you. We have reviewed all the values mentioned in the text. The difference between Figures 2Sa and 2Sc is that the first shows the RMSA, which represents the mean amplitude over a time window (in this case, 5 minutes), while the second shows the filtered data. The RMSA highlights sustained (long-lasting) events while attenuating individual transient explosions.

*Section 3.3 lacks the figure mentioned in the text.*

Yes, we are not sure why the figure was missing. We have now added it back, showing the tilt reconstructed with the seismometer.

**Citation**: https://doi.org/10.5194/egusphere-2024-3773-RC2

- **RC3**: 'Comment on egusphere-2024-3773', Anonymous Referee #3, 22 Jan 2025 reply

**Review on Geophysical fingerprint of the 4-11 July 2024 eruptive activity at Stromboli volcano, Italy**

This manuscript presents an extensive study on multi-parameter geophysical data such as seismic, acoustics, and ground deformation during the eruption of Stromboli volcano on 4-11 July 2024. The manuscript gives a good contribution in the field of volcano monitoring and also falls in the scope of NHESS. I think there are several points need to be improved before this manuscript could be accepted for publication.

1. The authors gave a detail and complete information in the introduction of the paroxysmal eruptions of Stromboli volcano, Italy, as well as the preceding eruptive activities and emphasizing the need of multi-parameter geophysical data in the eruption real-time monitoring. However, I found some parts in the introduction to be too extensive. It might be better to move the part of the conceptual models and other geological studies (line 65-76) into the discussion section instead, since the mentioned studies already focus on discussing the possible mechanisms generating paroxysmal eruptions in Stromboli.

2. In Figure 1b, there is no explanation on what N and CS refer to. While later on the text it is mentioned that N refers to the north crater area, it should also be mentioned in the caption of the figure.

3. Wrong station naming in line 128, ISTR3 instead of IST3 as the correct name I presume.

4. The authors used terminology seismic tremor and RMS as interchangeable terms e.g. in the caption of Figure 3a, these two terms have different meaning and could lead into confusion. It is also not clear in the text how the authors defined tremor in their observation. Did they use a threshold in the RMS seismic amplitude to separate tremor from the background energy? Was tremor present continuously all the time on 1-18 July or it started and stopped several time?

5. Figure 4c is too crowded to show the examples of the recorded LP events. I suggest to plot in a shorter time window to show a clear example of one or few recorded LP waveforms.

6. Figure of tilt derived from the seismic data is missing.

7. The authors did not write how they derived tilt from seismic data and only wrote the citations from the former publications which used the same method. For the completeness of the paper, this step needs to be included.

8. Figure 7 is missing.

9. The format used for date throughout the text is not uniform, for instance 4 July, 2024 on line 18 and July 7 on line 19. Please use the uniform date format throughout the manuscript.

10. I would prefer to separate the discussion into several sub-sections, for example separating the discussion of early eruption until the major explosion on 4 July, and then the paroxysmal eruption on 11 July. The interpretation of the observation and possible model from the literature studies could be discussed in each respective sub-section. This way, the discussion could become more focus and easier to follow.

REPLY

Many thanks for these comments. Below, you will find replies to each comment.

*1. The authors gave a detail and complete information in the introduction of the paroxysmal eruptions of Stromboli volcano, Italy, as well as the preceding eruptive activities and emphasizing the need of multi-parameter geophysical data in the eruption real-time monitoring. However, I found some parts in the introduction to be too extensive. It might be better to move the part of the conceptual models and other geological studies (line 65-76) into the discussion section instead, since the mentioned studies already focus on discussing the possible mechanisms generating paroxysmal eruptions in Stromboli.*

We agree with your comment. We have smoothed and refined the introduction, particularly the paragraph on the models. Some aspects of the model are further discussed in the main text, as noted in our response to comment 10. Thank you.

*2. In Figure 1b, there is no explanation on what N and CS refer to. While later on the text it is mentioned that N refers to the north crater area, it should also be mentioned in the caption of the figure.*

We have modified the caption. Thank you.

*3. Wrong station naming in line 128, ISTR3 instead of IST3 as the correct name I presume.*

Done, thank you.

*4. The authors used terminology seismic tremor and RMS as interchangeable terms e.g. in the caption of Figure 3a, these two terms have different meaning and could lead into confusion. It is also not clear in the text how the authors defined tremor in their observation. Did they use a threshold in the RMS seismic amplitude to separate tremor from the background energy? Was tremor present continuously all the time on 1-18 July or it started and stopped several time?*

Yes, the terminology was probably misleading. We have updated the text to use 'RMS tremor amplitude.' There is no threshold to separate the tremor from the background energy; we simply calculated the RMS within the tremor frequency range (1-3 Hz). Therefore, the tremor was recorded throughout the period, but with varying amplitudes, as explained in the text.

*5. Figure 4c is too crowded to show the examples of the recorded LP events. I suggest to plot in a shorter time window to show a clear example of one or few recorded LP waveforms.*

We have modified the figure showing just 20 minutes of July 3, where multiple LP events occurred. Thank you.

*6. Figure of tilt derived from the seismic data is missing.*

Yes, it's missing. We have added it.

*7. The authors did not write how they derived tilt from seismic data and only wrote the citations from the former publications which used the same method. For the completeness of the paper, this step needs to be included.*

We agree. We have rephrased the sentence adding more information that explains briefly how it's working: 'The relationship between displacement and tilt sensitivities is a function of the long-period corner frequency of the seismometer used. By applying the magnification factor (e.g., Aoyama et al. (2008), Genco and Ripepe (2010), and De Angelis and Bodin (2012)), which is constant around the natural period of the seismometer, we were able to convert the seismometer's output from displacement to ground tilt'.

*8. Figure 7 is missing.*

We have added it.

*9. The format used for date throughout the text is not uniform, for instance 4 July, 2024 on line 18 and July 7 on line 19. Please use the uniform date format throughout the manuscript.*

Thank you for this comment. We have standardized the text by adopting the journal's guidelines, using the format 'day month, year' and 'day month'.

*10. I would prefer to separate the discussion into several sub-sections, for example separating the discussion of early eruption until the major explosion on 4 July, and then the paroxysmal eruption on 11 July. The interpretation of the observation and possible model from the literature studies could be discussed in each respective sub-section. This way, the discussion could become more focus and easier to follow.*

We agree with this comment. We have divided the discussion into three subsections following a brief introduction: the first week of July, the second week of July, and the morphological changes in the crater area caused by the explosive activity. Each subsection discusses the eruptive models from the literature. Thank you.

**Citation**: https://doi.org/10.5194/egusphere-2024-3773-RC3

[revised manuscript text omitted]